

**Estimating hydraulic conductivity of a crusted loamy soil from beerkan experiments in a**
**Mediterranean vineyard**
Vincenzo Alagna[1], Vincenzo Bagarello[1], Simone Di Prima[2]*, Fabio Guaitoli[3], Massimo Iovino[1], Saskia
Keesstra[4,5], Artemi Cerdà[4,6]
[1] Department of Agricultural and Forest Sciences, University of Palermo, Viale delle Scienze, 90128 Palermo, Italy
[2] Agricultural Department, University of Sassari, Viale Italia, 39, 07100 Sassari, Italy
[3] Assessorato regionale dell'Agricoltura, dello Sviluppo Rurale e della Pesca Mediterranea, UO S5.05, Viale Regione Siciliana 2771, 90145 Palermo
Italy
[4] Soil Physics and Land Management Group, Wageningen University, Droevendaalsesteeg 4, 6708PB Wageningen, The Netherlands.
[5] Civil, Surveying and Environmental Engineering, The University of Newcastle, Callaghan 2308, Australia.
[6] Department of Geography, University of Valencia, Blasco Ibáñez, 28, 46010 València, Spain
* Corresponding Author. E-mail: sdiprima@uniss.it
**Abstract**
In bare soils of semi-arid areas, surface crusting is a rather common phenomenon due to the impact of
raindrops. Water infiltration measurements under ponding conditions constitute a common way for an
approximate characterization of crusted soils. In this study, the impact of crusting on soil hydraulic
conductivity was assessed in a Mediterranean vineyard (western Sicily, Italy) under conventional tillage. The
BEST (Beerkan Estimation of Soil Transfer parameters) algorithm was applied to the infiltration data to
obtain the hydraulic conductivity of crusted and uncrusted soils. Soil hydraulic conductivity was found to
vary during the year and also spatially (i.e., rows vs. inter-rows) due to crusting, tillage and vegetation cover.
A 55 mm rainfall event resulted in a decrease of the saturated soil hydraulic conductivity, $K_s$, by a factor
close to two in the inter-row areas, due to the formation of a crusted layer at the surface. The same rainfall
event did not determine a $K_s$ reduction in the row areas (i.e., $K_s$ reduced by a non-significant factor of 1.05)
because the vegetation cover intercepted the raindrops and therefore prevented alteration of the soil surface.
The developed ring insertion methodology on crusted soil, implying pre-moistening through the periphery of
the sampled surface, together with the very small insertion depth of the ring (0.01 m) prevented visible
fractures. Consequently, beerkan tests carried out along and between the vine-rows and data analysis by the
BEST algorithm allowed to assess crusting-dependent reductions in hydraulic conductivity with
extemporaneous measurements alone. Testing the beerkan infiltration run in other crusted soils and

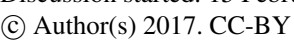



establishing comparisons with other experimental methodologies appear advisable to increase confidence on
the reliability of the method, that seems suitable to allow simple characterization of crusted soils.


**Keywords:** Hydraulic conductivity, water infiltration measurements, soil surface crust, vineyard, BEST
procedure

**1. Introduction**
The impact of raindrops on a bare soil surface can result in physical and chemical changes of the exposed
soils. The mechanical alteration of the upper soil aggregates, expressed in terms of compaction, splash and
particle detachment, contribute to form a surface crust (Assouline, 2004). This type of crust, named structural
crusts, differ from depositional crusts (West et al., 1992), which are formed by deposition of detached, fine
particles carried out in suspension by runoff (Fox and Le Bissonnais, 1998). The hydraulic properties of
crusts vary significantly (Fox et al., 1998a, 1998b). Different physical rainfall properties may be related with
structural crust development, such as intensity (Baumhardt et al., 1990; Freebairn et al., 1991; Morin and
Benyamini, 1977), kinetic energy (Eigel and Moore, 1983; Mohammed and Kohl, 1987) and momentum
(Brodie and Rosewell, 2007; Rose, 1960). The initial or wetting phase in crust formation is defined as
surface sealing (Römkens, 1979). During the drying cycle, this layer consolidates and may differ from the
wetting phase in its mechanical and hydraulic properties (Mualem et al., 1990). This drying phase is known
as crusting (Römkens, 1979).
The hydrodynamic properties of such a layered system (crust layer, underlying soil) may severely affect
the partition between infiltration and runoff at the soil surface, especially in arid and semi-arid areas where
crusting is a common phenomenon (Angulo-Jaramillo et al., 2016). Water infiltration measurements
constitute a common way for an indirect characterization of sealed/crusted soils (Alagna et al., 2013;
Bedaiwy, 2008). The Beerkan Estimation of Soil Transfer (BEST) parameters procedure developed by
Lassabatere et al. (2006) is a very attractive method for practical use since it allows an estimation of both the
soil water retention and hydraulic conductivity functions. The BEST method focuses specifically on the van
Genuchten (1980) relationship for the water retention curve with the Burdine (1953) condition and the



Brooks and Corey (1964) relationship for hydraulic conductivity. BEST estimates shape parameters, which
are texture dependent, from particle-size analysis by physical-empirical pedotransfer functions, and scale
parameters from beerkan experiments (Haverkamp et al., 1996), i.e. three-dimensional (3D) field infiltration
experiments at ideally zero pressure head. BEST substantially facilitates the hydraulic characterization of
unsaturated soils, and it is gaining popularity in soil science (Bagarello et al., 2014a; Castellini et al., 2016;
Di Prima, 2015; Di Prima et al., 2016b; Gonzalez-Sosa et al., 2010; Mubarak et al., 2010). Alternative
algorithms, i.e., BEST-slope (Lassabatere et al., 2006), BEST-intercept (Yilmaz et al., 2010) and BEST-
steady (Bagarello et al., 2014b), and field procedures based on BEST method were developed (Alagna et al.,
2016; Bagarello et al., 2014c; Di Prima et al., 2016a). The ability of the BEST method to distinguish between
crusted and non-crusted soils was demonstrated by Souza et al. (2014). Moreover, Di Prima et al. (2016a)
successfully applied a beerkan experiment involving different heights of water pouring on the infiltration
surface to explain surface runoff and sealing generation phenomena occurring during intense rainfall events.
These authors concluded that if any seal forms at the surface, the beerkan infiltration test should detect its
impact on flow and BEST estimates should essentially indicate the hydraulic properties of the surface layer.
In fact, the BEST method was developed for non-layered soils that are assumed to be uniform and have a
uniform soil water content at the beginning of the infiltration run (Lassabatere et al., 2006, 2009) and should
not contain a macropore network (Lassabatere et al., 2014). However, completely homogeneous soils are
very rare in natural environments (Reynolds and Elrick, 2002). Therefore, the hydraulic conductivity
obtained by an infiltrometer method, such as BEST, should probably be considered as an equivalent
conductivity, i.e. the conductivity of a rigid, homogeneous and isotropic porous medium characterized by
infiltration rates that are the same as those actually measured on the real soil (Bagarello et al., 2010). For the
case of stratified media, the layer with the lowest hydraulic conductivity generally controls the flow and
consequently cumulative infiltration at the surface (Alagna et al., 2013). Therefore, water infiltration data
can be regarded as representative of the hydraulic behavior of the least permeable layer, and therefore the
derived BEST parameters can be assigned to this layer. This approach was proposed by Lassabatere et al.
(2010) for a stratified medium with a low permeability sedimentary layer at the surface, by Yilmaz et al.
(2010, 2013), for the characterization of crusted reactive materials, and, recently, by Coutinho et al. (2016)
for a permeable pavement for stormwater management in an urban area.



In this paper we tested the BEST method in an agricultural setting with general objective to carry out a
hydraulic characterization of a loamy soil in a vineyard under conventional tillage located at Marsala
(western Sicily, Italy). In particular, both row and inter-row areas were sampled since a crust layer only
developed in the latter area. Therefore, the specific objective was to check the ability of the BEST method to
yield plausible estimates of saturated hydraulic conductivity of crusted and non-crusted soils.

**2.   Material and methods**
**2.1.   Study site**
The experimental site is located close to Marsala (western Sicily, Italy), in the homeland of Sicilian
viticulture (37°48'5.10" N and 12°30'44.79" E). Elevation is 111 m a.s.l. and soil surface is flat. The soil is a
typic Rhodoxeralf with a depth of 1 m and a small amount of gravel. According to the USDA classification,
the soil texture, determined on two replicated soil samples, is loam (**Table 1**). A weather station is located 5
km away from the sampling site (37°79'35.64"N and 12°56'81.59"E). It is positioned at the same elevation as
the sampling site and it is part of a network of stations managed by Servizio Informativo Agrometeorologico
Siciliano –SIAS.
At the sampling site, the common soil management for the vineyards of Marsala was applied during the
two years of sampling (2015 and 2016) (**Figure 1**). The soil is tilled to a depth of 0.10-0.15 m in October,
after the first autumn rainfalls. Faba bean (*Vicia faba* L. var. *minor*) is sown in November between the rows.
In March, the legume biomass is cut and immediately incorporated into the soil with a rotary tiller to a depth
of 0.20 m. Finally, a new rotary tillage is performed in May and, only for the second year, this was also done
in June. This soil management practice is applied between the rows. Along the rows, a mechanical topper is
used at each soil tillage date to a depth of 0.10 m.

**2.2.   Soil sampling**
An area of approximately 100 m$^2$ was sampled on three different sampling campaigns covering two
growing seasons. The first two campaigns were carried out at the beginning and the end of September 2015,
respectively, and the third campaign was performed at the beginning of July 2016. Between the first two
sampling campaigns, the soil was not tilled and a total rainfall of 55 mm fell (**Figure 1**), which is



approximately 10% of the average annual precipitation for the area. In particular, a 29-mm event occurred
during the morning of 9 September, with a maximum recorded intensity of 25 mm h$^{-1}$. During the same day,
a total of 44.6 mm of precipitation was recorded. This rainfall led to the development of a weak but clearly
detectable surface crust (thickness of ~4 mm) (**Figure 2**). This phenomenon was only observed between the
rows and not along the rows. The second sampling was done one week after the last rainfall event. Finally, a
third sampling campaign was carried out during the following dry season in order to sample the soil after the
ordinary tillage practices and with moisture conditions comparable to the first sampling date.
On each sampling date, a total of 10 undisturbed soil cores (5 cm in height by 5 cm in diameter) were
collected at the soil surface close to the points where the infiltration tests were performed, 5 along the rows
and 5 between the rows. These cores were used to determine the dry soil bulk density, $\rho_b$ (g cm$^{-3}$), and the
soil water content at the time of the experiment, $\theta_i$ (cm$^3$ cm$^{-3}$). The soil porosity was calculated from the $\rho_b$
data, assuming a soil particle density of 2.65 g cm$^{-3}$. A disturbed soil sample (0–10-cm depth), collected both
along and between the rows, was used to determine the particle size distribution (PSD), using conventional
methods (Gee and Bauder, 1986). Fine size fractions were determined by the hydrometer method, whereas
the coarse fractions were obtained by mechanical dry sieving. The clay, silt and sand percentages were
determined from the measured PSD according to the USDA standards.

**2.3. Beerkan experiments**
For each sampling date, an area of approximately 100 m$^2$ was chosen and 14 beerkan infiltration runs
(Lassabatere et al., 2006) were carried out using a 15 cm inner-diameter ring. Seven runs were carried out
along the rows and seven on the bare inter-rows area (**Figure 3**). The steel ring was positioned between two
vine stocks along the row and in the same orthogonal direction between the rows. The ring was inserted to a
depth of about 0.01 m into the soil surface to avoid lateral loss of the ponded water. On crusted soil, to
prevent fracture of the upper layer during ring insertion, the soil outside the hedge of the ring was moistened
with 5 cm$^3$ of water by means of a syringe before insertion. After ten minutes, the ring was carefully inserted
to the pre-established short depth applying a slight pressure and a gentle rotation. This site preparation was
essential to prevent crust surface perturbation.





According to the guidelines by Lassabatère et al. (2006), for each run a known volume of water (150 mL)
was poured in the cylinder at the start of the experiment and the elapsed time during its infiltration was
measured. When the amount of water had completely infiltrated, another identical volume of water was
poured on the confined infiltration surface and the time needed for the complete infiltration was logged. The
procedure was repeated 15 times for each run by applying water at a small distance (3 cm of height) from the
infiltration surface. As is commonly suggested in practical application of a ponding infiltration method, the
energy of the water due to the application was dissipated on the fingers of a hand in order to minimize soil
disturbance (Reynolds, 2008).
Di Prima et al. (2016b) showed that all BEST algorithms, i.e. BEST-slope, BEST-intercept and BEST-
steady, led to similar results in most cases. However, BEST-slope appeared to yield more accurate estimates,
especially of the saturated soil hydraulic conductivity, $K_s$ (mm h$^{-1}$), but it was affected by a failure rate
higher than others algorithms (Bagarello et al., 2014b). In this study, such a problem did not occur and,
therefore, the BEST-slope algorithm (Lassabatere et al., 2006) was considered to estimate the whole set of
parameters of the hydraulic conductivity function. BEST focuses specifically on the Brook and Corey (1964)
relationship:
$$\frac{K(\theta)}{K_s} = \left( \frac{\theta - \theta_r}{\theta_s - \theta_r} \right)^{\eta} \tag{1}$$

where $K$ (L T$^{-1}$) is the soil hydraulic conductivity, $\theta$ (cm$^3$cm$^{-3}$) is the volumetric soil water content, $\theta_r$
(cm$^3$cm$^{-3}$) is the residual volumetric soil water content, $\theta_s$ (cm$^3$cm$^{-3}$) is the saturated volumetric soil water
content, and $\eta$ is a shape parameter linked to the soil textural properties. In BEST, $\eta$ is estimated from the
analysis of the PSD with the pedotransfer function included in the procedure, whereas $\theta_s$, $\theta_r$ and $K_s$ are scale
parameters. BEST considers $\theta_r$ to be zero, and $\theta_s$ was assumed to coincide with soil porosity in this
investigation, as suggested by many authors (Bagarello et al., 2011; Di Prima, 2015; Di Prima et al., 2016a;
Mubarak et al., 2010; Xu et al., 2009). In particular, Di Prima et al. (2016a) demonstrated that the assumed
coincidence between saturated soil water content and porosity did not practically affect the $K_s$ estimation.
BEST-slope estimates sorptivity, $S$ (mm h$^{-0.5}$), by fitting the experimental cumulative infiltration data on
the explicit transient two-term equation by Haverkamp et al. (1994):

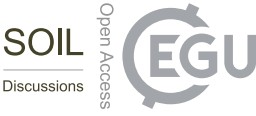

$$I(t) = S\sqrt{t} + \left[ A(1-B)S^2 + B\, i_s \right] t \qquad\qquad (2)$$
where $I$ (mm) is 3D cumulative infiltration and $t$ (h) is the time. Then, $K_s$ (mm h$^{-1}$)is estimated as a
function of $S$ as follow:
$$K_s = i_s - AS^2 \qquad\qquad (3)$$
where $i_s$ (mm h$^{-1}$) is the experimental steady-state infiltration rate, which is estimated by linear regression
analysis of the last data points describing steady-state conditions on the $I$ vs. $t$ plot and corresponds to the
slope of the regression line. The constants $A$ (mm$^{-1}$) and $B$ can be defined for the specific case of a Brooks
and Corey relation (Eq. 1) and taking into account initial soil water content, $\theta_i$ (cm$^3$cm$^{-3}$), as (Haverkamp et
al., 1994):
$$A = \frac{\gamma}{r(\theta_s - \theta_i)} \qquad\qquad (4a)$$
$$B = \frac{2-\beta}{3}\left[ 1 - \left( \frac{\theta_i}{\theta_s} \right)^{\eta} \right] + \left( \frac{\theta_i}{\theta_s} \right)^{\eta} \qquad\qquad (4b)$$
where $\gamma$ (parameter for geometrical correction of the infiltration front shape) and $\beta$ are coefficients that
are commonly set at 0.75 and 0.6 for $\theta_i < 0.25\ \theta_s$, and $r$ (mm) is the radius of the source.

**2.4. Data analysis**
Data sets were summarized by calculating the mean, $M$, and the associated coefficient of variation, $CV$. In
particular, the $cl$, $si$, $sa$, $\rho_b$, $\theta_s$ values were considered site specific and therefore they were determined only in
duplicate ($cl$, $si$, $sa$, $N = 2$) or, considering their low variability ($\rho_b$, $\theta_s$), the arithmetic mean and the
associated $CV$ were calculated (**Table 1**). Temporal variability of $\theta_i$ was determined on the basis of ten
replicate samples on each sampling date (**Table 2**). The $K_s$ data were assumed to be log-normally distributed
since the statistical distribution of these data is generally log-normal (Lee et al., 1985; Warrick, 1998). The
geometric mean and the associated $CV$ were therefore calculated to summarize $K_s$ values using the
appropriate ''log-normal equations'' (Lee et al., 1985). Statistical comparison between two sets of data was
conducted using two-tailed t-tests, whereas the Tukey Honestly Significant Difference test was applied to
compare three sets of data. The ln-transformed $K_s$ data were used in the statistical comparison. A probability
level, $P = 0.05$, was used for all statistical analyses.

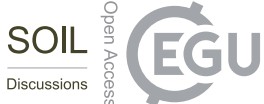


## 3. Results and discussion

196 In this paper, the BEST method was applied in an agricultural setting. In particular, the hydraulic

197 properties of a loamy soil were determined in a vineyard under conventional tillage located at Marsala

198 (western Sicily, Italy). The investigation was specifically aimed at checking the ability of the BEST method

199 to yield plausible estimates of saturated hydraulic conductivity of crusted and non-crusted soils, since a

200 limited experimental information is still available in the scientific literature (Souza et al., 2014).

201 Consequently, both row and inter-row areas were sampled since a crust layer developed only in the latter

202 portion of the field site. The 42 infiltrations runs were analyzed with the BEST-slope algorithm, yielding

203 positive $K_s$ values in all cases. In addition, the fitting of the infiltration model to the transient phase of the

204 infiltration run always yielded relative errors lower than 5.5% (Lassabatere et al., 2006), denoting an

205 acceptable error for transient cumulative infiltration (**Figure 4**).

206

### 3.1. Impact of surface crusting on hydraulic conductivity in vineyards

208 During the second field campaign, the crust layer only affected water infiltration between the rows (**Table**

209 **3**), suggesting that the protective role of vegetation along the rows was effective. The cover intercepted

210 raindrop energy preventing surface sealing (Dunne et al., 1991). The protective role along the vine-rows is

211 well known, while in vine inter-rows the mulching practice is commonly applied to protect soil from

212 raindrop impact (Celette et al., 2008; Prosdocimi et al., 2016). For the second campaign, the mean $K_s$ value

213 obtained between the rows was 1.6 times lower than the one obtained along the rows (**Figure 5**). In

214 particular, this latter value, equal to 212.4 mm h$^{-1}$, did not significantly differ from those of the first and third

215 sampling dates (**Table 3**). On the other hand, during these last two campaigns, beerkan runs carried out along

216 and between the rows also yielded similar $K_s$ values, due to the absence of a crust between the rows. This

217 experimental information suggested that the crusting occurrence, the adopted soil management and the cover

218 influenced both the temporal and the spatial variation of the soil hydrological characteristics at the field-

219 scale.

220 Bradford et al. (1987) reported for 20 soils (varying in texture from sand to clay) a reduction in

221 infiltration rate after 60 min of simulated rainfall (intensity of 63 mm h$^{-1}$), due to the effect of surface sealing





on infiltration. Bagarello et al. (2014c), Alagna et al. (2016) and Di Prima et al. (2016a) applied on five soils
having different texture a BEST derived procedure to explain surface runoff and disturbance phenomena at
the soil surface occurring during intense rainfall events. These authors reported saturated hydraulic
conductivity values of the disturbed soil from nine to 33 times lower than the undisturbed soils. In particular,
Di Prima et al. (2016a) applied this methodology in a vineyard with a sandy-loam texture. These authors
compared this simple methodology with rainfall simulation experiments establishing a physical link between
the two methodologies through the kinetic energy of the rainfall and the gravitational potential energy of the
water used for the beerkan runs. They also indirectly demonstrated the occurrence of a certain degree of
compaction and mechanical breakdown using a mini disk infiltrometer (Decagon, 2014). With this device,
they reported a reduction of the unsaturated hydraulic conductivity by 2.3 times due to the seal formation. In
another investigation carried out in Brazil with the BEST procedure, non-crusted soils were three times more
conductive than the crusted soil (Souza et al., 2014).


**3.2. Seasonal dynamics in hydraulic conductivity**

For the first and the third campaign, the beerkan runs carried out between the rows yielded comparable

and statistically similar (**Table 3**) $K_s$ values (**Figure 5**). In both cases, the average $K_s$ values were
approximately 20 times higher than the expected saturated conductivity on the basis of the soil textural
characteristics alone (e.g., $K_s$ = 10.4 mm h$^{-1}$ for a loam soil according to Carsel and Parrish, 1988). This
circumstance suggested that soil macroporosity generated by soil tillage in the ploughed horizon likely
influenced measurement of $K_s$ (Alagna et al., 2016; Di Prima et al., 2016a; Josa et al., 2010). In these
conditions, the soil structure is expected to be particularly fragile, especially with reference to macroporosity,
and hence unstable (Jarvis et al., 2008), which implies that clogging of the largest pores at the soil surface, as
a consequence of the aggregates breakdown occurring during a rainstorm, can easily mitigate tillage effects
on soil hydraulic properties (Ciollaro and Lamaddalena, 1998).

As discussed in the former section, the presence of the crust layer during the second field campaign

clearly affected water infiltration between the rows. In particular, the presence of this layer implied that $K_s$
was 1.5-1.8 times lower than that measured in the absence of the crusted layer (**Figure 5**). Crusting at the



soil surface determined an increased hydraulic resistance to water penetration into soil (Alagna et al., 2013)
since differences between the $K_s$ datasets (second against first and third sampling campaigns) were
statistically significant. Crusting also resulted in a decrease of the lowest measurable $K_s$ values, while the
highest values remained unchanged (**Table 3**).

The tillage practices carried out in the spring 2016 removed any existing soil crust and thereby increased

soil infiltration properties (**Figure 5**) (Chahinian et al., 2006; Ndiaye et al., 2005; Pare et al., 2011; Strudley
et al., 2008; Xu and Mermoud, 2001; Zhai et al., 1990), suggesting a cycling occurrence of crusting
phenomena within the year.

Many studies in the literature have reported similar dynamics, even in vineyards. In fact, infiltration

experiments constitute an indirect measurement closely associated with sealing or crusting (Römkens et al.,
1990), and the saturated hydraulic conductivity may vary considerably during the year if these phenomena
occur. In particular, rainfall and wetting–drying cycles favor soil reconsolidation and soil-surface sealing or
crusting, whereas tillage removes existing layering (Pare et al., 2011). For instance, Biddoccu et al. (2017)
studied temporal variability of soil hydraulic properties in a vineyard on a silt loam soil. These authors
reported hydraulic conductivity values measured during the summer four times lower than those measured
during the wet season, due to the presence of a structural crust resulting from rainfall events following the
late spring tillage.

**3.3. Applicability of the beerkan runs for the assessment of the crusted soil**

The results reported in this investigation were in agreement with those by Souza et al. (2014) and

therefore the supported the suggestion that the beerkan-based methodology should be usable to distinguish
between crusted and non-crusted soils.

Indeed, the hydrodynamic properties of both the crust and the underlying soil play a key role during a

rainstorm, affecting the partition between infiltration and runoff (Assouline and Mualem, 2002, 2006).
However, transient methods, as the beerkan one, appears appropriate to characterize crusted soils, since the
properties of the surface layer play a major role at early stages of the infiltration process (Vandervaere et al.,
1997). Recently, Di Prima et al. (2016b) showed that BEST-slope is less sensitive to the attainment of
steady-state and allows to obtain accurate estimates of saturated soil hydraulic conductivity with less water





and hence shorter experimental times than the other two BEST algorithms. For these reasons, BEST-slope
appears suitable, among the alternative algorithms, to characterize a crust layer. The applied methodology in
this investigation seems suitable to explore in the future the functional dynamics of the crust layer under
natural rainfall conditions.
A perplexity on the possibility to collect reliable data on crusted soils by a ponding infiltration experiment
is related to the need to insert the ring into the soil. The doubt is that ring insertion could determine fractures
in the crusted layer and these fractures could directly connect the ponded depth of water during the run with
the underlying, non-crusted, soil layer (Vandervaere et al., 1997). In other terms, ring insertion could
impede, in practice, measurement of fluxes though the crusted layer. In this investigation, fractures were not
visually detected at the soil surface, perhaps because the soil was not very dry when the experiment on the
crusted layer was performed (**Table 2**), the ring insertion depth was small (0.01 m), and insertion was carried
out a few minutes after moistening the insertion circumference. Other ponding infiltration techniques, such
as the single-ring pressure infiltrometer (Reynolds and Elrick, 1990) or, particularly, the simplified falling
head technique (Bagarello et al., 2004), presuppose appreciably deeper insertions of the ring and,
consequently, more risk to disrupt or alter the fragile crust layer at the soil surface during ring insertion.
Therefore, the beerkan run seems a more appropriate ponding infiltration run to prevent, or minimize,
substantial alteration of the surface to be sampled. Obviously, this conclusion needs additional testing but the
premises are encouraging, also considering that beerkan runs were successfully conducted in other crusted
soils (Souza et al., 2014).

**4.  Conclusions**
A loam soil was sampled in a Mediterranean vineyard located at Marsala (western Sicily, Italy), with
beerkan infiltration experiments carried out along the rows direction and in the inter-rows within two
consecutive growing seasons. Beerkan tests along with BEST-slope algorithm led to accurate estimates in
both crusted and un-crusted conditions, allowing to assess the effect of the cycling occurrence of crusting
due to rainfalls and wetting–drying cycles on the vineyard inter-rows.
A sampling strategy implying beerkan tests carried out along and between the vine-rows was successfully
applied. This strategy allowed to assess the reduction in hydraulic conductivity with extemporaneous



measurements alone. Its main advantage is that it allows a rapid assessment of crusting severity affecting
water infiltration. At the sampled site, the impact of crusting on saturated soil hydraulic conductivity was
moderate.
In conclusion, the hypothesis that the beerkan runs are suitable enough to detect the effect of the crust on
flow and BEST estimates appeared reasonable. In the future, the beerkan-based methodology should be
checked in other crusted soils. Comparisons should also be established with other experimental
methodologies.

**Acknowledgements**
This study was supported by grants from the Research Project CISV (grant n° 2014COMM-0363 CUP
872114000570002). A.V. carried out the experimental work. S.D.P. analyzed the results. All authors
contributed to write the paper. S.D.P. also thanks I.A., E.B. and R.D.O.

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

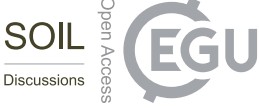



**Figure 1**. Precipitation and soil management program during the study period. The sapling dates are
reported.

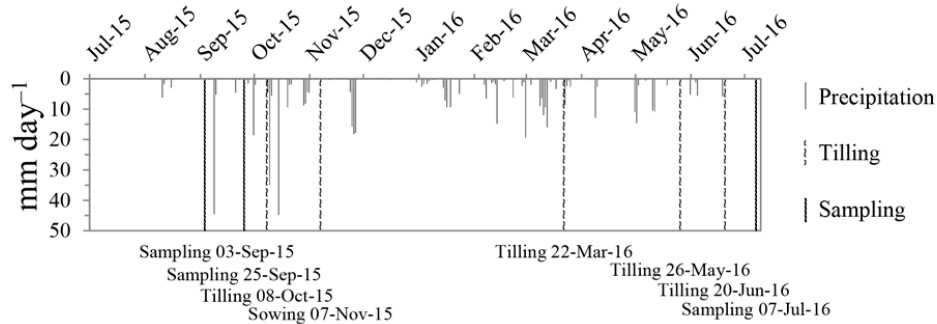








**Figure 2**. Surface crust layer developed after the intense storms fallen in September 2015.

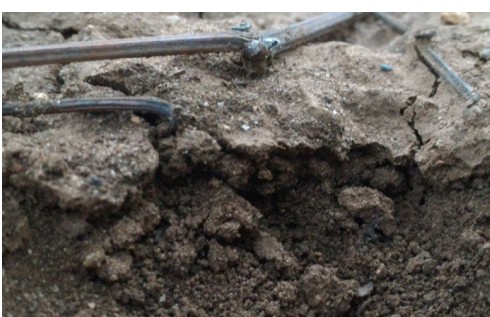


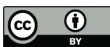

**Figure 3**. Beerkan infiltration runs carried out **(a)** along the rows and **(b)** on the bare inter-rows area.

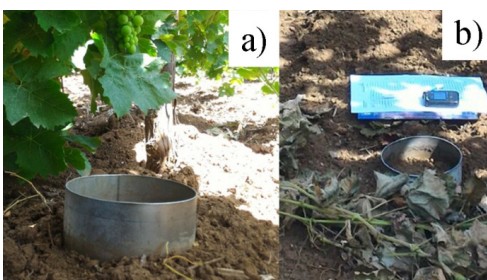




**Figure 4**. Cumulative frequency distribution of the relative errors, $E_r$ (%), of the fitting of the infiltration
model to the transient phase of the infiltration runs.

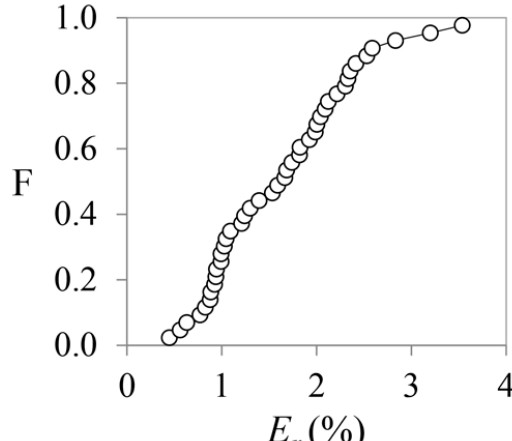





**Figure 5**. Box plots of the saturated soil hydraulic conductivity, $K_s$ (mm h$^{-1}$), values obtained from BEST
experiments carried out along and between the rows on different sampling dates and for different initial soil
water content, $\theta_i$ (cm$^3$cm$^{-3}$), values. On the box plots, boundaries indicates median, 25th and 75th quantiles,
the top and bottom whiskers indicate the minimum and maximum values.

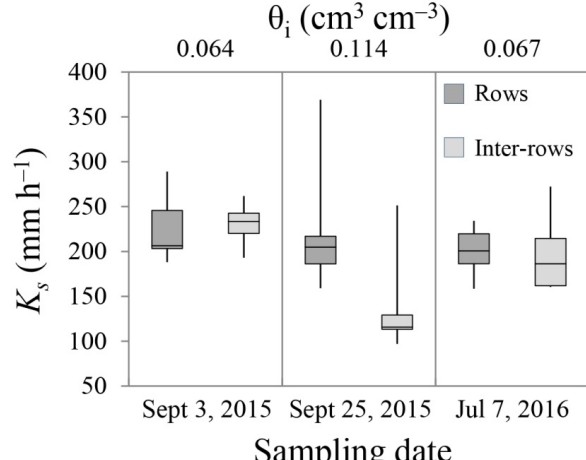





**Table 1**. Clay (%), silt (%) and sand (%) content (USDA classification system), soil textural classification,
dry soil bulk density, $\rho_b$ (g cm$^{-3}$), and saturated soil water content, $\theta_s$ (cm$^3$cm$^{-3}$), of the sampled vineyard
soil. Coefficient of variation (%) in brackets.

| Variable | Site characteristic |
|---|---|
| clay | 19.7 |
| silt | 49.6 |
| sand | 30.7 |
| Textural classification | loam |
| $\rho_b$ | 1.128 (5.1) |
| $\theta_s$ | 0.575 (3.8) |





**Table 2**. Sample size (N), minimum (Min), maximum (Max), mean, and coefficient of variation (CV, in %)
of the soil water content at the time of sampling, $\theta_i$ (cm$^3$cm$^{-3}$), values for different sampling dates.

| Statistic | Sept 3, 2015 | Sept 25, 2015 | Jul 7, 2016 |
|---|---|---|---|
| N | 10 | 10 | 10 |
| Min | 0.051 | 0.093 | 0.047 |
| Max | 0.073 | 0.133 | 0.087 |
| Mean | 0.064 A | 0.114 B | 0.067 A |
| CV | 12.0 | 10.9 | 18.1 |


The values in a row followed by the same upper case letter were not significantly different according to the
Tukey Honestly Significant Difference test ($P = 0.05$). The values followed by a different upper case letter
were significantly different.





**Table 3**. Sample size (N), minimum(Min), maximum (Max),mean, and coefficient of variation (CV, in %) of
the saturated soil hydraulic conductivity, $K_s$ (mm h$^{-1}$), values obtained from BEST experiments carried out
along and within the rows on different sampling dates.

| Variable | Position | Statistic | Sept 3, 2015 | Sept 25, 2015 | Jul 7, 2016 |
|---|---|---|---|---|---|
| $K_s$ | Rows | N | 7 | 7 | 7 |
| | | Min | 188.1 | 159.1 | 158.4 |
| | | Max | 289.1 | 369.1 | 234.2 |
| | | Mean | 223.6 a A | 212.4 a A | 199.2 a A |
| | | CV | 15.4 | 27.6 | 14.1 |
| | Inter-rows | N | 7 | 7 | 7 |
| | | Min | 193.0 | 97.0 | 160.6 |
| | | Max | 261.8 | 251.3 | 272.3 |
| | | Mean | 229.5 a A | 129.3 b B | 192.5 a A |
| | | CV | 10.3 | 31.7 | 20.2 |


The values in the column followed by a different lower case letter were significantly different according to a
two tailed t test ($P = 0.05$). The values in a row followed by a different upper case letter were significantly
different according to the Tukey Honestly Significant Difference test ($P = 0.05$).