# Peer review of "Estimating hydraulic conductivity of a crusted loamy soil from beerkan experiments in a"

_SOIL, 2016_

## Short Comment (SC1) · 12 Apr 2017

Dear authors,

I like the subject of soil crusting and I also like the way you wrote your paper. However, from my point of view you should extend and improve the data interpretation because at the moment the paper is too poor, I miss new scientific information. You could develop it by (i) implementing the data into hydraulic models in order to predict the hydraulic functions vs. time. Secondly I suggest to combine the rainfall intensity (=energy) information (which you have) with the change of the hydraulic conductivity. Some of these aspects have been done in the past. In order to stimulate doing more with your results, I recommand to read the following paper:

C.H. Roth et al. 1999: Impact of Tillage and Field Traffic Induced Variability of Soil Hydraulic Properties on Unsaturated Conductivity and Water Balance Calculations, 489-497. In: Proceedings of the International Workshop (1997) on Characterization and Measurement of the Hydraulic Properties of Unsaturated Porous Media, Part 2, edited by M. Th. van Genuchten, F.J. Leij and L. Wu. Published by the University of California, Riverside. Secondly I add a second paper to stimulate your interpretations.

Critique: Table 3 and Figure 5 are showing the same!

Please also note the supplement to this comment:
http://www.soil-discuss.net/soil-2016-79/soil-2016-79-SC1-supplement.pdf

---

## Author Comment (AC1) · 19 Apr 2017

Dear Prof. Gerd Wessolek,

First of all, we wish to thank You for Your comments, Your encouraging words and Your suggestion to improve the manuscript.
The specific objective of our investigation was to check the ability of the BEST method to yield plausible estimates of saturated hydraulic conductivity of crusted and non-crusted soils because the potential of the beerkan runs to detect the effect of the crust layer on flow is still largely unknown. There are some encouraging signs at this purpose but only a few investigations have been carried out. In a stratified

porous medium, the crusted layer may control the infiltration process and, following Lassabatere et al. (2010), the hypothesis was that BEST parameters may be assigned to the less permeable layer.

Two different approaches were applied in this investigation to check the ability of the BEST procedure to yield a different information between crusted and non-crusted soil. The first approach considered temporal changes of $K_s$ values obtained between the rows before and after the intense storms fallen in September 2015 that led to the development of a weak but clearly detectable surface crust. With this approach, the third field campaign allowed us to detect the restoring of higher $K_s$ values because tillage removed the crust. **Figure 1a** depicts the soil hydraulic conductivity functions obtained from averaged parameters for the different sampling dates at the vine inter-row area. While the curves of the first and the third sampling dates are close to one another, the curve of the second campaign departs for them rather clearly. Indeed, this result could be viewed as a suggestion that this latter curve might represent a crust layer characteristic curve. The second approach considered the spatial variation (i.e., rows vs. inter-rows) of conductivity, i.e. taking into account the protective role of the vegetation. A shift between the hydraulic conductivity functions for the row and inter-row areas was also detected, with the soil of the latter area denoting a reduced ability to conduct water for a given soil water content value (**Figure 1b**). Therefore, both approaches suggested that the BEST procedure was appropriate to show the impact of crusting on soil hydraulic conductivity.

However, we agree with You that the interest for our investigation could increase if we were able to show what are the hydrological implications of different soil hydraulic properties between the surface and the subsurface layers (Roth, 1997). At this purpose, we carried out some numerical simulations during the review of the manuscript with the objective to check the plausibility of our assumption that a beerkan infiltration run on the crusted soil layer is appropriate to hydraulically characterize this layer.

Numerical simulations were carried out using the graphical software packageVS2DI (Healy, 1990; Healy and Ronan, 1996), developed by the U.S. Geological Survey
for simulating the movement of water and transport of solute or heat in variably saturated porous media. In particular, the finite-difference method was used to solve the Richards equation for water flow. According to Nasta et al. (2012), a zero pressure head boundary condition was imposed on the soil surface delimited by the ring, while free drainage was set at the bottom of the modeled domain. The BEST derived parameters were used in VS2DI to obtain the simulated cumulative infiltration curves. In particular, the parameters estimated along the rows and on the bare inter-rows area during the second field campaign were used to define the hydraulic properties of the underlying soil and the crust layer (thickness of 4 mm), respectively. Moreover, with the aim to assess the impact of the hydraulic conductivity of the crust layer on the simulated curves, different $K_s$ values of the crust layer (from 93.6 to 165.6 mm h$^{-1}$) were considered. Then, the simulated cumulative infiltration curves ($I_{SIMULATION}$) were compared with the experimental BEST curve obtained on the crusted soil during the second field campaign ($I_{BEST}$ for $N$ data points at time $t_k$ using the root mean squared errors ($RMSE$s):

$$RMSE = \sqrt{\frac{\sum_{k=1}^{N}[I_{SIMULATION}(t_k)-I_{BEST}(t_k)]^2}{N}} \qquad (1)$$

**Figure 2** shows the relationship between $RMSE$ and $K_s$ values of the surface crust. The smallest deviation was obtained for a crust layer having $K_s$ = 133.2 mm h$^{-1}$. This value differed by a negligible factor of 1.03 from the in situ $K_s$ value obtained on the layered system (crust layer, underlying soil). This result supported our hypothesis that the experimental $K_s$ value was representative of the hydraulic behavior of the least permeable layer (i.e., the crust layer). Therefore, the derived BEST parameters could be properly assigned to this layer, which controlled the flow and consequently cumulative infiltration of the stratified medium.

Regarding your second suggestion, intensity and kinetic energy of the rainfall

were found to play a major role in determining mechanical changes of the soil surface (Baumhardt et al., 1990; Eigel and Moore, 1983). In the subsection 2.2. of the manuscript we reported some characteristics of the rainfalls occurred between the first and second sampling campaigns, which led the development of the surface crust.

Lastly, we agree that **Table 3** and **Figure 5** show redundant information and we will delete **Figure 5**.

**Reference**

Baumhardt, R. L., Römkens, M. J. M., Whisler, F. D. and Parlange, J.-Y.: Modeling infiltration into a sealing soil, Water Resour. Res., 26(10), 2497–2505, doi:10.1029/WR026i010p02497, 1990.

Eigel, J. D. and Moore, I.: Effect of rainfall energy on infiltration into a bare soil, 1983.

Healy, R. W.: Simulation of solute transport in variably saturated porous media with supplemental information on modifications to the US Geological Survey's computer program VS2D, USGS Numbered Series, U.S. Geological Survey ; Books and Open-File Reports Section [distributor],. [online] Available from: http://pubs.er.usgs.gov/publication/wri904025, 1990.

Healy, R. W. and Ronan, A. D.: Documentation of computer program VS2Dh for simulation of energy transport in variably saturated porous media; modification of the US Geological Survey's computer program VS2DT, USGS Numbered Series, U.S. Geological Survey : Branch of Information Services [distributor],. [online] Available from: http://pubs.er.usgs.gov/publication/wri964230, 1996.

Nasta, P., Lassabatere, L., Kandelous, M. M., Šimŭnek, J. and Angulo-Jaramillo, R.: Analysis of the role of tortuosity and infiltration constants in the Beerkan method, Soil Science Society of America Journal, 76(6), 1999–2005, 2012.

Roth, C. H.: Bulk density of surface crusts: depth functions and relationships to texture, Catena, 29(3–4), 223–237, 1997.

Best regards

Dr Simone Di Prima

On behalf of the authors

**Fig. 1.** Soil hydraulic conductivity functions obtained from averaged parameters for **(a)** different sampling dates and **(b)** along and between the vine-rows.

**Fig. 2.** **(a)** Root mean square errors ($RMSE$s) between the simulated and the experimental BEST curves vs. the saturated soil hydraulic conductivity, $K_s$, values of the surface crust. **(b)** Infiltration curve simulated considering a $K_s$ value of the surface crust equal to 133.2 mm h$^{-1}$ (line) compared with the measured BEST curve at the second field campaign in the inter-row area (dashed line). **(c)** Soil water content profile simulated with $K_s$ = 133.2 mm h$^{-1}$.

[Figure]

**Fig. 1.**

a)

b)

c)

**Fig. 2.**

[Figure]

---

## Referee Comment (RC1) · Anonymous Referee #1 · 1 May 2017

This paper presents the application of the beercan method to determine hydraulic properties of a crusted and non-crusted soil in a vineyard. The main conclusion is that method is an appropriate and practical method to characterize hydraulic properties of a crusted soil. Practical field methods are indeed very much needed to determine the effect of soil management on crust formation and its effect on soil hydrology. In that respect, this paper makes an interesting incremental contribution. Yet, the impact of the study would have been larger if the method was compared with another method. The conclusion that the beercan method gives reliable results cannot be checked since there were no other methods used against which the results obtained from the beercan method could have been compared. I also think that the presented results could

be extended by including other parameters that were determined from particle size distributions and used to estimate parameters and that were estimated from the beer-can method. Especially the soil sorptivity and related to it the mean pore size are parameters that were derived from the infiltration test but were neither presented nor discussed.

Ln 155: 'the BEST-slope algorithm (Lassabatere et al., 2006) was considered to estimate the whole set of parameters of the hydraulic conductivity function.' Only derived Ks values are given. If I understand the procedure correctly, then the shape parameters of the water retention curve (except for the parameter hg which is related (inversely proportional) to the mean pore size) were derived from the particle size distributions. Since the particle size distributions do not change over time, do not differ between the row and interrow, it can be assumed that these shape parameters are the same for the different measurement locations and times. However, I think that the authors should include the shape parameters that they used. The assumption that the texture does not vary is a reasonable assumption. But, I think that the authors also need to elaborate on an assumption that they implicitly make, namely that the shape parameters do not change when the structure of the soil change. Can it be assumed for instance that the shape parameters for a crust are the same as those for the loose soil? It can be argued that the shape parameters represent the hydraulic properties of a larger soil volume and that these are hardly affected by a thin surface crust layer. But, then it is important to explain why this assumption can be made for the shape parameters but not for the hydraulic conductivity. Besides the shape parameters, I think it is also crucial to include the estimated sorptivity and related to that the mean pore size parameter. I suppose that these parameters are derived from the data next to the saturated hydraulic conductivity. At first sight, one would expect that the hg is larger in the crusted soil because the pore size is smaller in the crusted soil. But, since the crust is very thin and the wetting front is below the crust, the capillary forces on the wetting front are exerted below the crust. Therefore, I expect that hg does not vary between the different locations and times. But it would be good to have this information. It is also necessary
to explain here why hg of the soil below the crust determines the infiltration rate and not hg of the crust. This is opposite to the control of the saturated conductivity of the crust on the infiltration rate.

Ln 215:'during the last two campaigns': I think the authors refer only to the last campaign on July, 7.

---

## Editor Comment (EC1) · K. Van Oost (Editor) · 23 May 2017

Thank you for the discussions. The first short comment posted will be considered as a review. The interactive discussion will be closed and based on the reviewers report, a recommendation will be made.